# Inkjet-printed transparent electrodes: Design, characterization, and initial in vivo evaluation for brain stimulation

**Rita Matta**[1◉]**, Davide Reato**[1,2◉]*****, Alberto Lombardini**[2]**, David Moreau**[1‡]*****,
**Rodney P. O'Connor**[1‡]*

**1** Mines Saint-Etienne, Centre CMP, Departement BEL, F - 13541 Gardanne, France, **2** Institut de Neurosciences de la Timone, UMR 7289, CNRS and Aix Marseille Université, 13005 Marseille, France

◉ These authors contributed equally to this work.
‡ DM and RPO also contributed equally to this work.
* davide.reato@univ-amu.fr (DR); david.moreau@emse.fr (DM); rodney.oconnor@emse.fr (RPO)

## Abstract

Electrical stimulation is a powerful tool for investigating and modulating brain activity, as well as for treating neurological disorders. However, understanding the precise effects of electrical stimulation on neural activity has been hindered by limitations in recording neuronal responses near the stimulating electrode, such as stimulation artifacts in electrophysiology or obstruction of the field of view in imaging. In this study, we introduce a novel stimulation device fabricated from conductive polymers that is transparent and therefore compatible with optical imaging techniques. The device is manufactured using a combination of microfabrication and inkjet printing techniques and is flexible, allowing better adherence to the brain's natural curvature. We characterized the electrical and optical properties of the electrodes, focusing on the trade-off between the maximum current that can be delivered and optical transmittance. We found that a 1 mm diameter, 350 nm thick PEDOT:PSS electrode could be used to apply a maximum current of 130 µA while maintaining 84% transmittance (approximately 50% under 2-photon imaging conditions). We then evaluated the electrode performance in the brain of an anesthetized mouse by measuring the electric field with a nearby recording electrode and found values up to 30 V/m. Finally, we combined experimental data with a finite-element model of the in vivo experimental setup to estimate the distribution of the electric field underneath the electrode in the mouse brain. Our findings indicate that the device can generate an electric field as high as 300 V/m directly beneath the electrode, demonstrating its potential for studying and manipulating neural activity using a range of electrical stimulation techniques relevant to human applications. Overall, this work presents a promising approach for developing versatile new tools to apply and study electrical brain stimulation.

## Introduction

Electrical brain stimulation is a long-established technique for studying brain function [1,2]. In addition to its research applications, electrical stimulation has been used clinically to treat

**Data availability statement:** Data to reproduce the figures in the manuscript is available on Figshare: https://doi.org/10.6084/m9.figshare.28302644.v1

**Funding:** This work was funded by a Marie Skłodowska-Curie postdoctoral fellowship (D.R.: HORIZON-MSCA-2021-PF-01 101063075), the European Union's Horizon 2020 research and innovation programme under grant agreement N°101034324 (D.R.), the French government under the France 2030 investment plan, as part of the Initiative d'Excellence d'Aix-Marseille Université – A*MIDEX AMX-22-COF-132 (D.R.), the French National Research Agency, through the «Investissements d'Avenir» program (ANR-21-ESRE-0003). The funders had no role in study design, data collection and analysis, decision to publish, or preparation of the manuscript.

or alleviate symptoms of various neurological conditions, either using intracranial or transcranial techniques [3–6], as well as for neuroprosthetic purposes [7,8].

Thanks to insights from computational models and in vitro and in vivo experiments, we now have a relatively precise understanding of the electric field distributions in the brain generated by different electrical stimulation techniques [9,10], as well as their effects on individual neurons [11–13]. The primary direct effect of electrical stimulation on neurons is a change in membrane potential [14,15], which depends on several factors, including the direction and magnitude of the electric field, the specific waveform used, and the morphology and electrophysiological properties of the cell [16–19]. If the electric field is sufficiently strong (>20 V/m), membrane voltage changes can trigger action potentials in quiescent neurons [17,20]. However, even lower field magnitudes, such as those generated in the brain during transcranial electrical stimulation techniques, can still modulate spike rate and timing [21–23].

Despite knowing how electrical stimulation affects single neurons, predicting its impact on population activity is challenging due to neuronal variability in morphology, electrophysiology, and synaptic properties [24–26]. This is a crucial problem because the capacity of electrical stimulation to modulate brain function depends on its global effects on brain tissue. From a practical standpoint, this issue also presents significant challenges in designing stimulation protocols that achieve the desired outcomes. Current approaches to modulate brain activity at the population level often involve matching the stimulation frequency to the overall temporal pattern of neuronal activity (as estimated from the local field potential or EEG signals) [27,28], using predefined stimulation parameters that have been shown to effectively modulate clinical symptoms in pathological conditions [29] or, more recently adaptive/closed-loop approaches to modulate more selectively brain activity [30–32]. While valuable, these approaches must be complemented by a deeper understanding of how electricity affects brain activity.

One potential approach to achieve that is to study the effects of electrical stimulation across multiple spatial scales, ranging from single neurons to populations. Electrophysiological extracellular recordings enable recording the activity of a large number of neurons but they are susceptible to electrical artifacts during stimulation. An alternative is to use 2-photon imaging (a microscopy technique capable of imaging deeper into living tissue than other methods) with fluorescent indicators that are sensitive to either the presence of calcium ions [33] or to changes in the membrane potential [34]. These techniques offer the potential to monitor the overall activity of entire regions using wide-field imaging, while also allowing the user to zoom in and monitor the activity of hundreds or thousands of neurons simultaneously, as well as individual neurons or specific cellular compartments [35]. Genetic tools that are readily available in mice enable these recordings to be performed in specific cell types [36], providing a valuable resource for understanding how electrical stimulation can affect different cells in the brain. Thus, this approach holds potential for the development of new stimulation paradigms that are better suited for specific applications.

A straightforward method for combining electrical stimulation with imaging techniques is to place the electrodes directly on the brain's surface (or dura mater). However, this approach may significantly limit the field of view, relying on using multiple small electrodes and imaging in the space between them, an approach that has been taken for recording purposes [37]. An alternative strategy is to use nanotubes, nanowires or nanomeshes because of the minimal alteration of the light path offered by these structures [38,39]. However, the biocompatibility of the metals often used in these structures has not yet been fully proven. Fabricating devices using transparent materials represents another option. For instance, indium-tin-oxide (ITO) has been used to fabricate transparent devices. However, this inorganic material is expensive, brittle and limited in terms of flexibility [40]. Graphene is an excellent alternative for

transparent electrodes [41–43] but cleanroom fabrication remains less cost-effective. Furthermore, the capacity for charge storage and injection with this material may be restricted [43].

Ideally, electrodes for brain stimulation and concurrent imaging should be highly biocompatible, transparent, and capable of injecting sufficient charge to generate electric fields comparable to those used in established human stimulation protocols. A material that meets the requirements is PEDOT:PSS (poly(3,4-ethylenedioxythiophene) polystyrene sulfonate). PEDOT:PSS is a conductive polymer that has been shown to be highly biocompatible and well-suited for recording electrical activity and stimulating brain tissue [44]. For instance, metal electrodes coated with PEDOT:PSS exhibit lower impedances, resulting in improved signal-to-noise ratio [45,46]. Coating metal electrodes with PEDOT:PSS also increases the amount of charge that can be stored and used for stimulation purposes, with an increasing thickness leading to higher charge injection capacity [47,48]. This feature is primarily due to the mixed ionic/electronic conductance of PEDOT:PSS material that confers volumetric capacitance to PEDOT:PSS films [49]. Recent work has also demonstrated that electrodes coated with PEDOT:PSS can induce neural responses in the mouse visual cortex for over a year, thus highlighting the high biocompatibility of this material [50].

In this study, we directly inkjet-print PEDOT:PSS to produce an electrode ideal for neurostimulation. Inkjet printing is a drop-on-demand patterning technique with micron precision [51] that allows rapid prototyping and testing while limiting wasted material. This technique is gaining popularity for producing devices that interface with biological tissue, with the goals of recording and stimulation [52]. One of the main advantages of using PEDOT:PSS inks is that their properties can be easily altered [53] to achieve films with desired electrical, mechanical properties while maintaining stable performances in water [54]. One example is a recent study [55] that proposed directly inkjet printing PEDOT:PSS electrodes onto the scalp of human subjects (rather than pre-manufacturing them [56]) for electroencephalography (EEG) applications, as this could improve electrical contact, which is often affected by the presence of hair [57]. Thin films of PEDOT:PSS can also be highly transparent [58,59]. Notably, a study [60] demonstrated that inkjet-printed PEDOT:PSS electrodes, using a large volume insulated except for a small electrolyte-contact area, can achieve transparency and ideal electrical properties for localized recording or stimulation. This is attributed to the small contact area combined with the material's volumetric capacitance. This work highlights the feasibility of inkjet printing PEDOT:PSS for fabricating electrodes for brain stimulation.

Here, we confirmed that inkjet-printed PEDOT:PSS electrodes can exhibit high charge injection capabilities while maintaining high transparency and we further characterized the tradeoff between transparency and charge injection capabilities as a function of electrode thickness. In addition, we evaluate their transparency under 2-photon imaging conditions and perform in vivo experiments in a mouse, combined with a finite-element model (a computational method for simulating physical phenomena by dividing a complex structure into smaller, simpler elements), to determine the maximum electric field that these electrodes can generate. Our findings indicate that the electrodes we fabricated can produce electric fields in the mouse brain up to 300 V/m, which meets the requirements of many electrical stimulation modalities. Therefore, inkjet-printed electrodes represent a promising new tool for studying the effects of electrical stimulation in combination with simultaneous imaging techniques.

## Results

We fabricated a device with a transparent electrode for stimulation using a hybrid approach that combines standard microfabrication techniques in a cleanroom environment with inkjet printing (Fig 1A). Inkjet printing enables the design and quick patterning of potentially

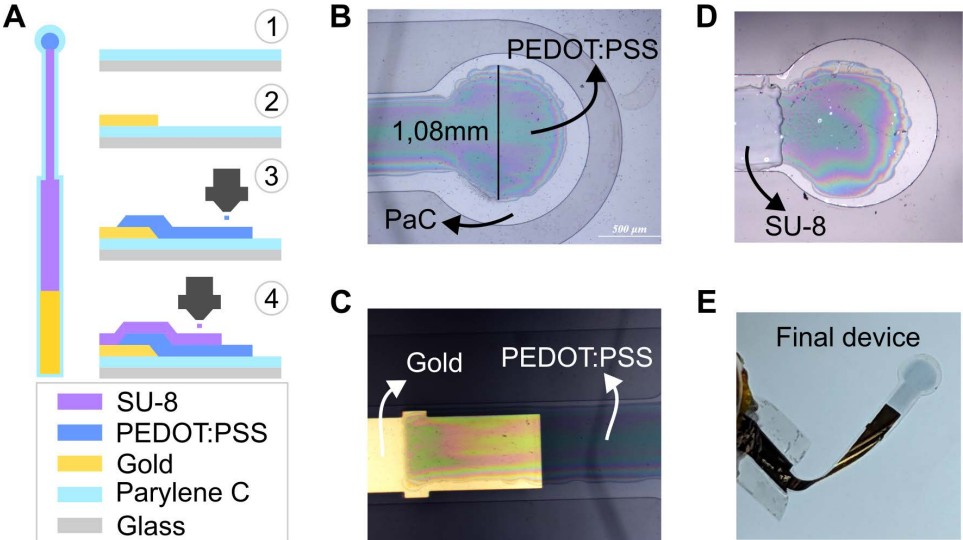

**Fig 1. Device fabrication.** A: Schematics of the device and the fabrication process. Vapor deposition, photolithography and gold deposition were used to deposit Parylene C and create the connection line. The electrode, a single circle with a 1 mm diameter, was inkjet printed using a PEDOT:PSS ink. Insulation was applied by printing multiple layers of the photoresist SU-8. B: Picture of a representative device after printing PEDOT:PSS. C: Interconnection between gold and the printed PEDOT:PSS layer. D: Picture of a representative device after printing 3 layers of SU-8. E: Example of released device ready for testing.

multi-layered materials with varying thicknesses on desired substrates. Our primary goals in this study are to leverage this process to print conductive polymer electrodes suitable for electrical stimulation, to investigate the relationship between their charge injection capacity and transparency and to test them in vivo.

Briefly, the fabrication process began with the chemical vapor deposition of Parylene C (PaC) on a glass substrate, to provide a layer for encapsulation and electrical insulation. Next, photolithography was used to pattern the device-to-generator connection, followed by the deposition of 120 nm of gold using thermal evaporation. After defining the device's outline via photolithography and etching (to facilitate the release from the substrate), we then inkjet printed PEDOT:PSS, using a custom ink formulation [54] to define the electrode. The electrode consists of a single circle with 1 mm diameter (Fig 1B, see *Discussion* for an explanation on the chosen size) and the ink was overlapped with the gold lines to create an electrical connection between the two (Fig 1C). A SU-8 ink was then printed to insulate the device leaving only the exposed electrode (Fig 1D). The device was subsequently removed from the substrate, resulting in a highly transparent electrode within a highly flexible device (Fig 1E). A more detailed explanation of the fabrication process is reported in the *Materials and Methods* as well as a full schematics of the process in S1 Fig. To explore the importance of thickness, we fabricated devices with varying numbers of printed PEDOT:PSS layers, thus controlling this parameter. This allowed us to estimate the electrical and optical properties across different devices.

## Electrical characterization

We began by measuring the overall thickness of the printed PEDOT:PSS layers, as this variable likely influences both the charge storage capacity of the electrodes (and thus the injectable current) and the device transparency. The overall thickness of the device was determined by

the number of printed PEDOT:PSS layers (Fig 2A). Electrodes made from a single layer had a thickness of 350 ± 63 nm (mean ± SD, n = 5 electrodes) with each additional layer further increasing the total thickness. For example, electrodes with four layers of PEDOT:PSS exceed 1 μm in thickness. While the volumetric capacitance of PEDOT:PSS suggests that adding more layers should enhance the maximum current injection capacity of electrodes, this increased thickness may compromise the device's transparency, potentially limiting its use in imaging applications. Here we will first characterize the electrical properties of the electrodes, followed by an analysis of their transparency in the subsequent section.

We characterized the electrical properties of the devices in phosphate-buffered saline (PBS). We first estimated the sheet resistance of printed PEDOT:PSS using four-point probe measurements (see *Materials and Methods*). We performed the measurements using both one layer and two layers. We estimated values of 346 ± 3 Ω/□ and 174 ± 6 Ω/□ ("ohms per square", mean ± SD, n = 7 measurements for one printed layer and n = 6 for two layers),

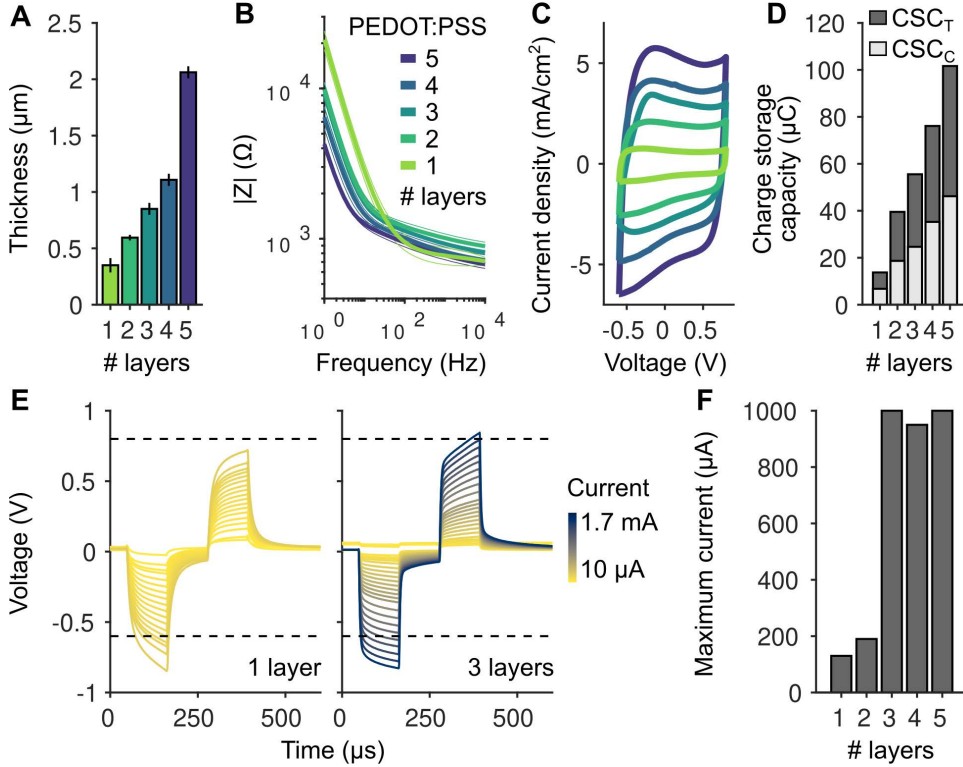

**Fig 2. Electrical properties of inkjet printed electrodes.** A: Thickness of electrodes made of a different number of printed PEDOT:PSS layers (n = 5, 3, 5, 2, 2 electrodes for 1-5 layers respectively). B: Electrical impedance spectroscopy measurements for electrodes made of different numbers of inkjet printed PEDOT:PSS layers. Thick lines represent mean values across electrodes and thin lines the error bars estimated as standard deviation (n = 9 electrodes for 1 printed layer and and n = 3 for the others). Due to their low impedance, the electrodes are compatible with commercially available electrical stimulators. C: Cyclic voltammetry (CV) measurements for electrodes made of different numbers of inkjet printed PEDOT:PSS layers. Larger areas enclosed by the cyclic voltammetry curves indicate higher charge storage capacity of the electrodes. D: Charge storage capacity (cathodic: light gray, total: dark gray) estimated from the cyclic voltammetry measurements (calculated from the areas under the cyclic voltammetry curves) for different numbers of layers. E: Example traces (for one and three layers PEDOT:PSS) from pulse tests to estimate the maximum current that can be applied using the electrodes. Dashed lines represent the water window ([-0.6 0.8] V), defining the voltage range where the electrode/electrolyte interface remains stable and avoids significant water electrolysis. F: Maximum current that can be injected using the electrodes before reaching the water window, beyond which undesirable side reactions may occur.

corresponding to conductivity values of 131 ± 1 S/cm and 133 ± 5 S/cm, consistent with values obtained with the same ink formulation [54] as well as with a study reporting approximately 100 S/cm for a similar, though not identical, ink formulation [61]. Note that the similar conductivity (their values are not significantly different, p = 0.43, t-test) is expected considering that the prints have different thickness but are made of the same material. Nevertheless, similar values confirm that there is no electrical barrier between two layers.

We next performed Electrochemical Impedance Spectroscopy (EIS) to determine the overall impedance of the electrodes as a function of the number of printed layers (Fig 2B). We found that the low-frequency impedance decreased with increasing number of printed layers while staying lower than 1 kΩ for frequencies higher than 1 kHz. At the lowest frequency tested (1 Hz), the impedance for one printed layer was just 21 ± 4 kΩ (mean ± SD, n = 9 electrodes), a low value that allows to use this electrode with a large set of commercially available stimulators.

We then estimated the amount of charge that the electrodes could store as a function of the number of layers using Cyclic Voltammetry (CV, Fig 2C). The CV, representing the current density over the potential, has approximately a rectangular shape. The absence of noticeable redox peaks implies no significant electrochemical reactions and instead indicates a double-layer charging. Charge storage capacity (CSC), which is the total charge transferred during the cyclic sweep, can be determined directly from the CV by integrating the current over the potential range (-0.6 to 0.8 V). Increasing the number of printed layers, and the resulting increased thickness of the electrode, leads to higher current storage capacity (Fig 2D), as expected considering the increased capacitance of the device.

Finally, we applied test pulses (Fig 2E) to identify current values, in either the anodic or cathodic phases, that lead to voltages outside the water window, the range of potentials where water does not undergo significant electrolysis. Operating outside this range can cause undesirable electrochemical reactions, such as gas evolution and pH changes. For our PEDOT:PSS electrodes, the water window is -0.6 V to 0.8 V when using a Ag/AgCl reference electrode. We found a value of max 130 μA for one layer electrodes before crossing the water window. This value increases to 190 μA for two layers and jumps abruptly to about 1 mA at higher numbers of layers (Fig 2F).

In summary, the results of our electrical characterization suggest that the electrical properties of the circle electrode can be tuned by altering the number of printed layers. In general, the impedance and maximum current that can be safely injected suggest that already one single layer of PEDOT:PSS can be used to inject a significant amount of current and the low impedance allows its use with regular commercial stimulators. Increasing the number or layers leads to higher maximum currents but the increased thickness of the device can become problematic because it leads to lower transparencies, a topic that we address in the following section.

## Transmittance measurements

We characterized the transparency of the devices to find a thickness that balances sufficient current injection with high transparency. We estimated the transparency of our devices using two distinct methods (Fig 3A). First, we employed a 1-photon spectrophotometer to measure the transmittance of multiple PEDOT:PSS printed layers. This method quantifies the proportion of light passing through the material relative to the initial incident light intensity (Fig 3A, left). Second, we sought to estimate transmittance under 2-photon excitation conditions (Fig 3A, right). In this setup, a pulsed near-infrared laser was used to excite molecules in the sample by simultaneous absorption of two photons, and the resulting fluorescent emission light was collected. The overall transmittance can then be estimated by comparing the measured

fluorescence signal in presence of the electrode to that obtained without the electrode. As both the excitation and emission light are attenuated by the electrode in the 2-photon setup, we anticipated lower transmittance values compared to the spectrophotometer measurements. Importantly, the spectrophotometer-derived transmittance values can be directly used to predict the transmittance under 2-photon conditions (see *Materials and Methods*). This allows for the generation of theoretical predictions, which we could test with a 2-photon microscope.

We first estimated the transmittance by printing 2x2 cm squares with varying numbers of PEDOT:PSS layers (1 to 5) and analyzing the amount of light passing through the material. These measurements were performed at distinct wavelengths using a one-photon stimulation spectrophotometer (Fig 3B; see *Materials and Methods*). Approximately 84% of light in the

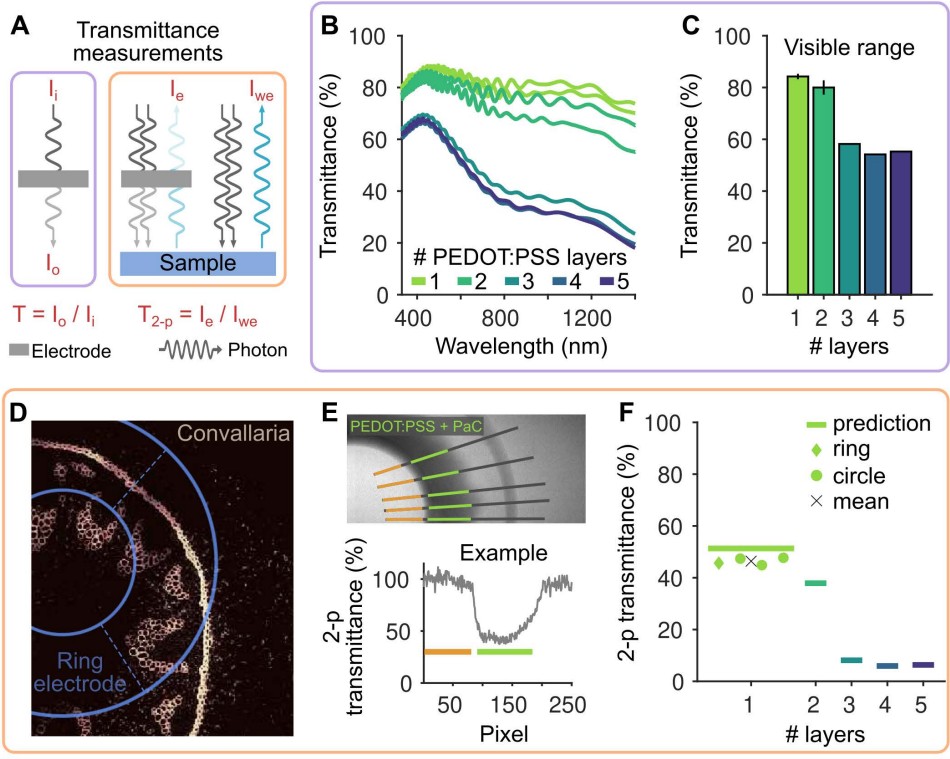

**Fig 3. Optical properties of inkjet printed electrodes.** A: Schematics of the tests performed to estimate the transmittance of the electrodes. One set of tests relied on measurements with a spectrophotometer (violet color, relative to panels B and C), which estimates how much light passes through the sample compared to the initial one ($I_i$: input light, $I_o$: output light). The other set of tests are in 2-photon conditions (orange color, relative to panels D-F), where two photons stimulate a sample and the following emission light is collected by the sensor. Transmittance can be estimated by comparing the emission light collected through the electrode ($I_e$) and without the electrode in the light path ($I_{we}$). B: Light transmittance as a function of light wavelength for an increasing number of printed PEDOT:PSS layers. C: Average transmittance in the visible range (380-750 nm) estimated from the data in B. Errorbar indicates mean and STD (n = 2, 2, 1, 1, 1 electrodes for 1-5 printed layers respectively). D: 2-photon imaging of a glass slide containing the fluorescent plant Convallaria majalis. A ring-like electrode made of a single layer of PEDOT:PSS is placed on the top of the slide (the position is drawn in blue). E: 2-photon experiment to estimate the transparency of electrodes made of a single printed layer of PEDOT:PSS. (Top) A ring electrode is placed on the top of a fluorescent glass slide. Manually selected lines for the analysis are indicated in gray while green indicate the location of the PEDOT:PSS + PaC electrode. Orange lines represent the baseline considered, where no electrode is present. (Bottom) Example of the estimated transmittance across one selected line (after correcting for the background). F: Estimated transparency in 2-photon imaging conditions of electrodes made of a single printed layer of PEDOT:PSS. Symbols represent the experimental measurements and black cross the mean values across electrodes. The solid lines represent the theoretically predicted transmittances derived from the data in panel B (see *Materials and Methods*).

visible range (380–750 nm) passes through a single layer of PEDOT:PSS (Fig 3C). Two printed layers of PEDOT:PSS exhibit an 80% transmittance, while further increasing the number of layers results in a rapid decrease in transmittance.

We then conducted tests under 2-photon imaging conditions. Initially, we placed a single-layer PEDOT:PSS electrode (ring-shaped to fit within a single field of view) on top of a glass slide containing a fluorescent plant (Convallaria majalis) to determine if fluorescent structures could be visualized beneath the electrode. Our results confirmed this visualization was possible (Fig 3D). To quantitatively estimate the transmittance of the electrodes, we imaged uniform fluorescent slides, which allowed us to directly quantify the emission light collected beneath the electrode compared to a region without the electrode in the light path. We extracted the light intensity over multiple manually drawn lines, identified the electrode location, and estimated the light loss under the electrode compared to a location without the electrode, after correcting for the background (Fig 3E; see also *Materials and Methods*). In the represented example, the transmittance of a single layer of PEDOT:PSS was just below 50%. This estimate was confirmed using multiple full-circle electrodes (47 ± 2%, mean ± SD across 3 electrodes, Fig 3F).

Based on the transmittance values obtained using the spectrophotometer, we computed the theoretical transmittance expected under 2-photon conditions (lines in Fig 3F), accounting for the multiple obstructions of the light path by the electrode (see *Materials and Methods*). The experimental values were slightly below the theoretical predictions, suggesting that spectrophotometer measurements alone can be used to derive accurate estimates of transmittance in 2-photon experiments. Notably, a measured transmittance of about 50% using a spectrophotometer, as measured for 3 to 5 layers, corresponds to only 5-10% transparency in 2-photon conditions, making it less suitable for imaging experiments. However, a two-layer electrode may only lose approximately 10% transmittance compared to a single-layer one.

## In vivo experiments and computational modeling

Considering that a single-layer PEDOT:PSS electrode can deliver currents exceeding 100 μA (Fig 2F) while preserving approximately 50% transmittance for 2-photon imaging (Fig 3F), we opted to use a single PEDOT:PSS printed layer for subsequent in vivo experiments, where we aimed to directly estimate the electric field generated in the brain by the electrodes.

The effects of electricity on brain activity depend on the electric field generated within the tissue [12]. Thus, we aimed to estimate the magnitude of the electric field produced by a single-layer PEDOT:PSS electrode placed on the dura mater of an anesthetized mouse (epidural stimulation). Estimating the electric field can be achieved by measuring voltage fluctuations induced by the stimulation at various locations within the brain. Thus, we recorded voltages in the mouse brain using rigid electrodes referenced to a screw placed in the back of the animal skull. For stimulation, we placed the transparent PEDOT:PSS electrode on the mouse dura and a metal screw counter electrode in the front part of the brain, to approximate a monopolar stimulation configuration (Fig 4A). We applied sinusoidal 10 Hz current–controlled stimulation at multiple amplitudes across multiple repetitions. We limited the maximum current to 100 μA, just below the maximum value that was identified during the device characterization, as higher currents resulted in the electrode losing functionality (see S2 Fig). The electrode was lowered perpendicularly to the brain surface, and voltage fluctuations induced by the stimulation were recorded at multiple depths (Fig 4B). We estimated the electric field component perpendicular to the brain surface by calculating the difference between the amplitudes of the measured sinusoidal voltages at consecutive depths and dividing by their distance. We found that the electric field magnitude scaled

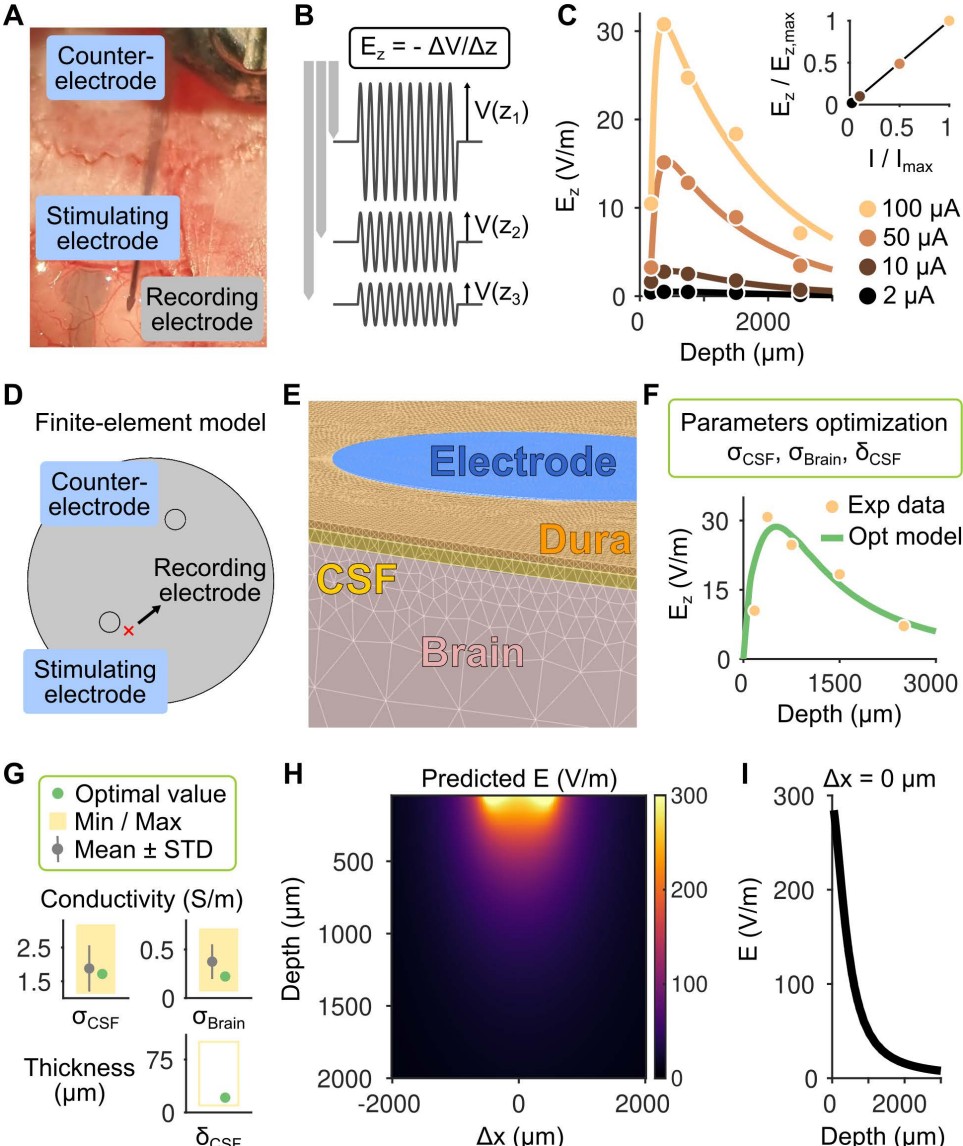

**Fig 4. Combined in vivo experiment and computational modeling to estimate the electric field generated by the inkjet-printed electrode.** A: The flexible printed PEDOT:PSS electrode is placed on the mouse brain for stimulation, together with a screw counter electrode in the front to close the circuit. A rigid electrode is used to record voltage fluctuations due to a sinusoidal current stimulation (10 Hz). B: The component of the electric field perpendicular to the brain surface ($E_z$) is estimated by taking the difference of the amplitude of the recorded sinusoidal voltages at multiple depths (schematic representation). Electric field measurements offer insights into the potential effects of the stimulation at specific spatial locations in the brain. C: Estimated electric field ($E_z$) for four different currents applied, indicated by different colors. Lines indicate a double exponential fit for each stimulation amplitude. The top plot shows the linearity between current applied and the electric field (both of which are normalized by the values corresponding to the maximum current applied). The black line is the identity line. D: Geometry used for the FEM to simulate the experimental conditions. The stimulating and counter electrodes match the geometry and location of the experiment. E: Representation of the FEM meshing with different colors representing different tissue types (electrode, dura, CSF, brain). Each voxel represents a volume element within the model, where equations governing current flow are solved to estimate the current distribution. F: Experimental data and simulation results for a model in which three parameters (the conductivities of the brain and CSF and the thickness of the CSF layer) were optimized. The model, with optimized parameters, accurately reproduces the experimental data. G: Comparison of the optimal parameters identified with the means and standard deviations derived from multiple studies and reported by Sim4Life (see *Materials and Methods*). Yellow rectangles represent the minimum and maximum values reported in the literature, which were used as the bounds for the optimization procedure. The CSF thickness varies depending on the location in the brain and therefore we left quite large boundaries for the optimization. The optimized parameters closely match those

reported in the literature. H: Predicted electric field beneath and close by the stimulating electrode. Colors indicate electric field magnitude as calculated by the model that best reproduces the experimental data. I: Dependence of the electric field magnitude on the depth directly below the center of the stimulation electrode. The electric field rapidly decays with increasing depth in the brain.

linearly with the applied current (slope of the fit, β = 0.999, R² = 0.9998; top plot in Fig 4C). At an applied current of 100 μA, the peak electric field was approximately 30 V/m (Fig 4C). This field strength could be sufficient to elicit action potentials in some neurons [17], as the minimum activation threshold is estimated to be around this value (see *Discussion*). Additionally, a measurement of 30 V/m approximately 1 mm from the center of the stimulating electrode suggests that the electric field directly beneath the electrode may be an order of magnitude larger.

To test this hypothesis, given the practical challenges of measuring voltage fluctuations directly beneath the electrode, we developed a computational approach where we numerically simulated our experimental conditions while optimizing the model parameters based on the experimental data. By doing so, we aimed to estimate the electric field distribution in regions where direct measurements were unfeasible. Specifically, we implemented a finite-element model (FEM) where we positioned the active electrode and counter-electrode as in the experimental setting and assigned them the same geometrical properties as in the experiments (Fig 4D, see *Materials and Methods*), an approach that is standard to simulate electrical brain stimulation [62–64]. We modeled the electric currents reaching the brain, passing through the dura mater and the cerebrospinal fluid (CSF) underneath, three tissue types that we defined in our simulation (Fig 4E). We applied 100 μA current to the simulated PEDOT:PSS electrode (as in the experiments) and set the counter electrode to ground to close the circuit. Rather than manually assigning electrical conductivities to the tissues, an optimization algorithm was used to determine the optimal set of parameters that best matched the experimental data (Fig 4F). This optimization process focused on the conductivities of the CSF and brain tissue, as well as the thickness of the CSF layer (the subarachnoid space), aiming to minimize the difference between the simulated and experimentally measured perpendicular component of the electric field at the corresponding location (see *Materials and Methods*). We found a set of parameters that led to an electric field that closely matched the data in the corresponding position (Fig 4F). We further checked whether the optimal parameters were consistent with the ones commonly used in the literature (see *Materials and Methods*) and found them to be within one standard deviation from the mean (Fig 4G), suggesting that our experimental measurements aligned with predictions from basic current flow across the tissues. The optimal value for the thickness of the CSF layer was also compatible with experimental data from previous reports [65]. We then analyzed the distribution of the electric field magnitude generated by the model in a full 2D space that included areas directly beneath the stimulating electrode (Fig 4H). We found that the electric field directly below the center point of the electrode reached approximately 300 V/m (Fig 4I), a value about 10 times the minimum threshold for direct neuronal activation [66].

In summary, by combining an in vivo experiment with a parameters-optimized FEM we were able to estimate the full distribution of the electric field inside the brain, including directly beneath the stimulating PEDOT:PSS electrode. This electric field distribution and magnitude could modulate large neuronal populations while minimizing variability in stimulation effects often associated with non-uniform field distributions. This advantage is particularly significant compared to multi-electrode stimulation approaches, which often generate highly non-uniform electric fields.

## Discussion

In this work, we showed that a ~ 1 mm diameter electrode made of a single PEDOT:PSS printed layer (thickness ~ 350 nm) allows a transmittance of approximately 50% in 2-photon imaging settings and can inject currents of up to ~ 130 μA, which correspond to an electric field magnitude of maximum 300 V/m in the mouse brain when delivered on the dura mater. These results were derived combining microfabrication, inkjet printing, electrical and optical device characterization, in vivo experiments and computational modeling with parameters optimization procedures.

A single, relatively large electrode design was chosen for two primary reasons. First, this configuration served as a proof-of-principle demonstration that inkjet-printed electrodes of this type are viable for neural stimulation. Second, minimizing electric field variations beneath the electrode is challenging with multi-electrode arrays. The large, single electrode design mitigates this challenge, enabling more uniform and controlled neuronal stimulation, a feature that is further enhanced by our direct estimation of the electric field generated by the electrode.

Our strategy for the fabrication of the device was hybrid combining standard cleanroom techniques with inkjet printing. This approach was chosen to facilitate the analysis of the printed part of the device without adding the complexity of potential issues that may arise from the connector part, as our cleanroom fabrication procedure is already well established. Future work will focus on developing fully printable devices [52], for instance by directly patterning the connectors with gold ink. This approach holds significant promise for advancing the development of flexible devices [67], by leveraging the cost-effective fabrication provided by printing techniques, an additive manufacturing method that minimizes waste and can be performed without a cleanroom environment.

In our study, we confirmed the results of Donaldson and Swisher [60] by showing that printed PEDOT:PSS can be used to deliver current. Our estimates indicate that the charge storage capacity of our 350 nm printed PEDOT:PSS electrodes is approximately 1 mC/cm$^2$ while Donaldson and Swisher reported 6 mC/cm$^2$ for their smallest volumes of PEDOT:PSS electrodes (300 μm outer diameter electrodes). Despite having a lower charge storage capacity, our electrodes may offer higher transmittance. Donaldson and Swisher reported transmittance values of 77% at 510 nm and 68% at 900 nm. Our experimental results using a two-photon microscope showed that the theoretical calculation that we used to relate the transmittance measured using a spectrophotometer to that under two-photon conditions is accurate. We can then predict the transmittance of the electrodes proposed by Donaldson and Swisher to be $0.68^2 * 0.77 * 100 = 35\%$, so about 10-15% lower than the one we measured using our thinnest electrodes. Thus, comparing our results with theirs highlights the trade-off between charge injection capabilities and transparency, a central theme of this work. This differs from Donaldson and Swisher's focus, which was on characterizing how varying the volume of PEDOT:PSS, while limiting the surface area in contact with the electrolyte, could yield electrodes with suitable electrical properties for recording or stimulation. Building upon their study, we also demonstrate the in vivo applicability of these electrodes and directly measure electric fields in their vicinity. Furthermore, we developed a finite-element model, with parameters fitted to experimental data, to achieve a full characterization of the electric field, including in the region beneath the electrode.

While in our study we used PEDOT:PSS, the predominant materials for transparent electrodes are graphene and sputtered metal oxides, such as ITO. ITO exhibits high transparency (~90%) and excellent conductivity (~$10^4$ S/cm), with a sheet resistance as low as 36 Ω/□ [68], but its inherent brittleness limits its flexibility. Conversely, graphene demonstrates over 90% transparency across the ultraviolet to infrared spectrum, coupled with high electrical ($10^4$ - $10^5$

S/cm) and thermal conductivity, as well as flexibility and biocompatibility [69]. Nevertheless, its high cost and complex synthesis processes may limit its broader use for electrode fabrication. PEDOT:PSS, in contrast, distinguishes itself by combining transparency with flexibility [70], while offering mixed electronic and ionic conductivity. Furthermore, it benefits from commercial availability and ease of processing and manipulation [48,71]. Despite its significant promise, PEDOT:PSS presents certain limitations, including restricted conductivity, flexibility, and mechanical stability [72]. The material's complex structure, which involves both electronic and ionic transport, makes its properties highly sensitive to processing techniques. Furthermore, the mechanical stability of PEDOT:PSS can be problematic, with potential issues such as cracking, delamination, and re-dispersion in aqueous environment [73]. While incorporating cross-linkers can enhance film stability, this often comes at the cost of reduced electronic conductivity [74].

Our approach presents several limitations. In applications requiring highly localized electric fields, smaller electrodes would be preferable. While our fabrication method has a demonstrated resolution of approximately 30 μm [54], enabling the printing of multiple stimulation electrodes, their individual charge injection capacities would be reduced. Our tests in PBS confirmed that increasing the number of printed PEDOT:PSS layers enhances the maximum injectable charge, but this comes at the cost of decreased electrode transparency. To achieve higher current densities in applications where transparency is critical or when smaller electrodes are necessary, several strategies could be considered. Modifications to the ink formulation [75], including alternative dopants or functionalization strategies [76], may improve charge injection without compromising transparency. Multi-layer coatings of silver nanowires/PEDOT:PSS films represent another promising approach to enhance charge injection while maintaining or even improving transparency [77]. Future studies should also address the challenge of reliable current delivery at elevated levels to mitigate delamination at the gold/PEDOT:PSS interface. Strategies to achieve effective current delivery at elevated levels may include surface treatments to enhance adhesion [78,79] or the incorporation of a buffer layer to reduce stress. Another limitation is the lack of long-term biocompatibility and functionality data for our electrodes. Although a recent study demonstrated that PEDOT:PSS electrodes can deliver low-frequency electric fields without cytotoxicity [80], and another in vivo study showed that PEDOT:PSS coated electrodes could elicit responses in the visual cortex of mice for over a year [50], the longevity of our specific electrode design in vivo remains to be determined. A longitudinal study in mice, for instance periodically measuring electrode impedance with and without current application, would be necessary to assess long-term performance. Furthermore, potential heating of the electrode and tissue during stimulation requires further investigation. Experiments using thermal cameras and varying current levels will be necessary to characterize these thermal effects. In addition, histological analysis will be needed to assess potential tissue damage, including markers of inflammation, cell death, or gliosis [81].

Despite these limitations, by combining in vivo experiments with a parameters-optimized FEM approach, we were able to estimate the electric field directly below the electrode. Finding that the optimal parameters for the fit are consistent with values reported in the literature validated our modeling approach and the subsequent use of the model to estimate the electric field directly below the electrode. While direct measurements could be performed to test the predictions, they would require insertion of electrodes at extreme angles to measure the most superficial layers of the cortex. Etching a small hole to allow for the insertion of a recording electrode directly below the stimulating PEDOT:PSS electrode would also greatly alter current flow and thus yield an altered estimate of the electric field. Nevertheless, the close agreement between the experimental data and the simulation results, obtained using physiologically plausible parameter values compatible with the literature, indicates that the fundamental

principles of current propagation adequately describe the observed experimental outcomes. This also confirms that this type of modeling predicts the actual measured electric fields in the brain with great accuracy [10].

Our results suggest that the electric field achieved under the electrode may be as high as 300 V/m. This raises the question of how this number compares to common applications of electrical brain stimulation. Different electrical brain stimulation modalities generate electric fields that span different orders of magnitudes. Invasive techniques, such as deep brain stimulation (DBS) or cortical microstimulation generate electric fields from hundreds to thousands of V/m depending on electrode size and distance from the electrode [62,82]. Electroporation techniques rely on electric fields in the range of $10^4$ to $10^7$ V/m, depending on the desired effects [83]. On the other hand, transcranial techniques, such as transcranial direct and alternating current stimulation (tDCS, tACS), generate electric fields at the cortical level of max about 1 V/m [10,63], although electroconvulsive therapy can generate fields (of short duration) of up to hundreds of V/m [84]. Temporal interference stimulation, another form of transcranial electrical stimulation, can generate electric fields of maximum 0.4 V/m in deeper areas in the brain [85,86]. Electric fields in the order of 100 V/m in the human cortex, generated using transcranial magnetic stimulation, can also induce motor responses [87,88]. In general, a threshold of 15-20 V/m is considered as the minimum to induce action potentials and 5 times that value is considered enough for robust activation [17,66].

Considering this evidence from the literature, our electrodes are well suited to study the effects of different electrical stimulation techniques on brain activity, to either induce spiking activity or modulate neuronal activity with lower amplitude electric fields, which are nevertheless effective in modulating neuronal activity [13,89]. For instance, while the effects of low-amplitude electrical stimulation on firing activity have been characterized for decades in vitro and in vivo using extracellular recordings [14,21–23,90–96] or calcium imaging [97–101], a direct estimation of the membrane voltage fluctuations induced by electrical stimulation have been mainly limited to somatic compartments [15,17,90,102,103] or estimated only in in vitro preparations [104]. With the rapid development of new indicators to directly monitor changes in membrane voltage in mice [34,105,106], our transparent electrode represents the ideal tool to test predictions on the direct effects of electrical stimulation on the neuronal membrane potential from decades of computational modeling work, including those from more recent years [18,107–109].

While further testing is required to confirm long-term biocompatibility and functionality, our device represents a significant advancement due to its cost-effectiveness and rapid testability. This makes it valuable not only for research but also for clinical applications requiring effective stimulation or recording, as well as optical access or imperceptibility [110]. Transparent PEDOT:PSS electrodes, for instance, are also used in biosensors and wearable health monitoring devices, offering conductivity without sacrificing flexibility or transparency [111]. Furthermore, advanced transparent systems like Carbon Layered Electrode Arrays (CLEAR), composed of graphene and Parylene C, have successfully integrated neuro-electrical signal recording with optical coherence tomography for cortical blood vessel monitoring [112]. This underscores the potential clinical relevance of transparent electrode technologies, including those based on PEDOT:PSS, for future applications.

Although direct tests on the effects of stimulation on brain activity are a priority and necessary, our study suggests the feasibility of using inkjet-printed PEDOT:PSS transparent electrodes for both neural stimulation and imaging. These electrodes offer a unique combination of transparency, flexibility, and conductivity, making them attractive for a range of biomedical applications. Future research will focus on enhancing their long-term stability, integrating them into multi-electrode arrays for complex stimulation patterns, and

developing fully printable devices with optimized ink formulations. These advancements could significantly impact neuroscience and bioelectronics research, potentially leading to new clinical tools.

## Materials and methods

### Device fabrication

The device was made using a combination of standard cleanroom techniques and inkjet printing and the detailed process is represented in S1 Fig.

Glass slides, which were used as the substrate, were sonicated in a 2% soap solution, then rinsed, and sonicated in acetone and isopropanol. A 3 μm layer of Parylene C (PaC) was deposited via chemical vapor deposition *(Specialty Coating Systems, PDS 2010, USA)*. Photolithography with AZ nLOF 2070 *(Microchemicals, Germany)* negative photoresist was employed to pattern the gold structures. This involved spin-coating (500 rpm for 10 seconds, and 3000 rpm for 40 seconds), soft baking (2 minutes at 110 °C), exposure (186 mJ/cm$^2$ at the i-line wavelength), post-exposure baking (2 minutes at 110 °C), and development in AZ 826 MIF developer. Titanium (10 nm) and gold (120 nm) layers were deposited via electron beam and thermal evaporation respectively, followed by a lift-off process involving acetone and isopropanol. A second photolithography step with AZ 10XT *(Microchemicals, Germany)* positive photoresist defined the device's outline. After spin-coating (3500 rpm for 35 seconds), soft baking (2 minutes at 110 °C), exposure (480 mJ/cm$^2$ at the h-line wavelength), and development in AZ developer, reactive ion etching (RIE) *(Oxford instruments, Plasmalab 80 + RIE, UK)* with $O_2$ and $CF_4$ gasses was employed to etch the PaC to define the outline.

A piezoelectric inkjet printer *(DMP 2800, Fujifilm Dimatix, Santa oven Clara, CA, USA)* patterned the PEDOT:PSS electrodes and the insulating SU-8 layer. Custom-formulated PEDOT:PSS ink was used based on previous work [54]. It includes commercially available PEDOT:PSS water dispersion *(Clevios PH1000 by Heraeus)*, DMSO *(Thermofisher, USA)* (10% w/w), IPA (5% w/w), TWEEN 20 (0.5% w/w), and GOPS (1% w/w). Further details on the ink formulation rationale and the mechanical and electrical properties of the PEDOT:PSS films are provided in [54]. The ink was mixed, sonicated, filtered, and printed with precise alignment. Printed layers were cured, and excess material was washed off. Samba cartridges with a 2.4 pL drop volume and 15 μm spacing patterned the PEDOT:PSS electrodes on the gold lines, then the prints were cured at 130 °C for 10 minutes. SU-8 2002 *(SU-8 2002; MicroChem Corp.)* was used for insulation due to its excellent chemical and mechanical properties. It was diluted with cyclopentanone for better printability and applied in three layers, followed by drying (95°C for 60 seconds) and UV cross-linking. The devices were finally released from the glass substrate by immersion in deionized water.

### Device electrical characterization

Sheet resistance was estimated using four-point probe measurements with a Jandel RM3000 unit *(Jandel, UK)*. Electrochemical impedance spectroscopy (EIS) was performed using a PalmSens4 potentiostat *(PalmSense BV, The Netherlands)* to analyze the electrical properties of PEDOT:PSS. Measurements were taken across frequencies from 1 to $10^6$ Hz in a two electrode setup with phosphate buffered saline (PBS). PEDOT:PSS served as the working electrode (WE), and silver-silver chloride (Ag/AgCl) was used as the counter and reference electrodes. A sinusoidal potential with a 10 mV amplitude was applied to the WE. The impedance spectrum exhibited capacitive behavior at lower frequencies and increasingly resistive behavior at higher frequencies. Increasing the number of PEDOT:PSS layers resulted in a noticeable decrease in impedance magnitude as seen in Fig 2B. Cyclic Voltammetry (CV) was used to

study the redox behavior of PEDOT:PSS in PBS by linearly sweeping the potential of the working electrode between -0.6 to 0.8 V at a scan rate of 1 V/s with steps of 1 mV. The cyclic voltammograms indicate no significant electrochemical reactions and thicker layers show larger currents, indicating enhanced capacitance (Fig 2C). The charge storage capacity (CSC), representing the total charge transferred during cyclic sweeps, was determined by integrating the current over the potential range (-0.6 to 0.8 V) from the cyclic voltammetry (CV) results. Increasing the thickness of PEDOT:PSS layers enhance the surface area available for charge storage, thereby increasing CSC as seen in Fig 2D. Charge Injection Capacity (CIC) was determined by applying biphasic current pulses (100 μs duration) *(Metrohm Autolab, Nova 2.1)* to electrodes made of different PEDOT:PSS layers and identifying the maximum current that could be used before crossing the water window (Figs 2E,F).

## Device transparency characterization

The transparency of the PEDOT:PSS electrodes was evaluated by assessing the transmittance of light through a printed square (2x2 cm) with various numbers of printed layers (1 to 5), using a spectrophotometer (*UV-2600, Shimadzu, Japan*). The instrument is equipped with a light source covering the specified wavelength $w$, ranging from 300 to 1400 nm. A monochromatic light passes through the sample (the printed squares), and the spectrophotometer measures the transmittance, which is the ratio of the intensity of the transmitted light to the intensity of the incident light. This measured transmittance $T(w)$ was used to predict the effective transparency for 2-photon experiments. By knowing the transmittance for the excitation laser ($T_{exc}$) and sample emission ($T_e$), we can provide an upper bound to the effective transmittance, $T_{2p} = T_{exc}^2 \cdot T_e$. The presence of the square factor is attributed to the two-photon excitation process. As the sample is excited by two photons, the excitation efficiency when crossing the electrode is reduced by a factor of $T_{exc}$ for each photon, resulting in an overall damping factor of $T_{exc}^2$. The wavelength of the stimulation light of the 2-photon microscope we used is 920 nm wavelength while we collect light at 520 nm. Thus, the transmittance is expected to be $T_{2-p} = T(920\,nm)^2 \cdot T(520\,nm)$. Considering that electrodes also contains a layer of PaC, whose transmittance we can assume to be uniform across visible light wavelengths and to be $T_{Pac} = 0.97$, the final transparency of the device should be no larger than $T_{2-p} = T(920\,nm)^2 \cdot T(520\,nm) \cdot T_{PaC}^3$. In Fig 3F, we show the expected transparency for different layers of PEDOT:PSS deposited on PaC.

Tests in 2-photon imaging conditions were performed using a 2-photon microscope (*Ultima 2P Plus, Bruker Corporation, MA, USA*). We acquired two images: one with the electrode in the field of view and one without. Using the image with the electrode, we extracted the intensity profile of five manually defined lines. These lines crossed the electrode and included a baseline region where there was no electrode (see an example in Fig 3E, top). After matching the baseline intensities of the signals extracted from the two images, we divided the signals from the image with the electrode by the signals from the image without. This normalized for the non-uniform background. As a result, we obtained a baseline defined at 100%. By measuring the average signal in the area where the electrode was present, we could directly estimate the transmittance in 2-photon conditions (see an example in Fig 3E, bottom). Overall transmittance for each tested electrode was defined as the average across the multiple lines we defined. The same analysis was performed for either ring-like or full circle electrodes. For one electrode, we could not image the full electrode within the microscope field of view and therefore we first compared the light loss from the PEDOT:PSS electrode to the PaC and then, using a separate image, from the PaC to an area with no device. We performed this two-step procedure using exactly the same approach described above.

## In vivo electrophysiology

All animal experimental procedures, aimed at minimizing any animal suffering during all surgical procedures, were approved and performed in accordance with the French Ministry of Higher Education, Research and Innovation (approval reference number APAFIS#22182–2019091818381938). Analgesic was administered intraperitoneal before starting the procedure (buprenorphine 0.3 mg/ml, diluted 1:10). Mice were then anesthetized with isoflurane in an induction box (4% volume in $O_2$) and subsequently placed in a stereotaxic frame (*Kopf Instruments*) where the anesthesia was maintained (1–1.5%). Eyes were covered with ointment and body temperature was controlled using a heating pad. After injecting lidocaine (20 mg/ml) subcutaneously and cleaning the skin with betadine, a midline incision was performed to expose and clean the skull. Two craniotomies were performed to insert two screws to use as a counter electrode for stimulation and a reference electrode for the recording system. A large craniotomy (3 mm in diameter) was performed between visual and somatosensory cortex and the flexible electrode was placed on the top of the dura mater (epidural stimulation). Neither conductive gel nor any other electrolyte was placed between the electrode and the dura. A small incision of the dura was performed to insert a rigid tungsten electrode insulated with PaC (#573220, *A-M Systems, WA, USA*) approximately 1 mm from the center of the stimulating electrode. Voltage signals were filtered and amplified (model 3000, *A-M Systems, WA, USA*) before being digitized at 1 kHz (NI DAQ USB-6001, *National Instruments, TX, USA*). The same DAQ was also used to output a 1 s, 10 Hz sinusoidal voltage that was converted to current by a linear stimulus isolator (*Soterix Medical Inc., NJ, USA*). The software to control the stimulus waveform and record voltage traces was written in MATLAB (*MathWorks Inc., MA, USA*). Recordings were performed at multiple depths and for multiple trials to improve the estimation of the voltages inside the brain. After the procedure, deeply anesthetized animals were sacrificed using cervical dislocation.

## Electric field estimation

To estimate the electric field the voltage signals were filtered (low-pass Butterworth filter, 20 Hz passband and 40 Hz stopband frequencies) and the peaks and troughs of the signal were identified and used to estimate the amplitude of the recorded sinusoids. By taking the median over the multiple stimulation cycles and over the multiple trials (20 repetitions) we were able to obtain a robust estimate of the amplitude of the recorded sinusoid. The normal component (perpendicular to the brain surface) of the electric field was computed by taking the opposite of the voltage differences across consecutive recording depths and dividing by their distance. The electric field values were spatially assigned to the midpoint between consecutive recording positions. To fit the electric field profile, we minimize the sum of the residuals between a double exponential function and the experimental points.

**Table 1. Minimum and maximum values of the parameters used for the optimization procedure.** Values to optimize are the conductivities of the CSF and the brain as well as the thickness of the CSF layer. Minimum and maximum values for the conductivities are from a review on estimations from the literature by Sim4Life (https://itis.swiss/virtual-population/tissue-properties/database/low-frequency-conductivity/).

| Parameter | Min value | Max value |
|---|---|---|
| $\sigma_{CSF}$ | 1.13 S/m | 3.19 S/m |
| $\sigma_{Brain}$ | 0.066 S/m | 0.717 S/m |
| $\delta_{CSF}$ | 10 μm | 100 μm |

## Finite element modeling and model fit to the data

We simulated our experimental configuration by setting up a FEM using COMSOL (*COM-SOL Inc., MA, USA*). Active and counter electrodes were placed at distances approximately as in the experiments. We simulated the brain as a cylinder with a layer of cerebrospinal fluid (CSF, the subarachnoid space) and dura mater above it. We simulated the PEDOT:PSS as a cylinder of 1 mm diameter and a thickness of 500 nm, while the counter as a cylinder with the same diameter but 200 μm thick. We set the conductance of the PEDOT:PSS to be $10^4$ S/m (consistent to our estimation during the device characterization) while the counter electrode, made of stainless steel to be $1.5 \cdot 10^6$ S/m. As in the experiments we applied a 100 μA inward current to the PEDOT:PSS electrode and set the counter electrode as ground. The electrodes were placed on the top of the dura mater to which we assigned a thickness of 20 μm [65] and a conductivity of 0.06 S/m. Note that the specific conductances of the electrodes do not have a strong influence in our simulations since we directly controlled the current as in the experiments (and the electrodes are not too thick). What is more critical to shape the electric field are the conductance ($\sigma_{CSF}$) and thickness of the CSF ($\delta_{CSF}$), and the conductance of the brain ($\sigma_{Brain}$). We decided to estimate these parameters based on our experimental data. Specifically, we implemented an optimization procedure to identify the parameters that lead to electric fields that best match our experimental recordings, namely the normal component of the electric field, $E_z$, that we recorded at a specific distance from the electrode. We decided to set up the optimization procedure considering not the experimental points directly but the double exponential fit that tracks the spatial profile of the electric field (as in Fig 4C). This greatly improves the efficiency of the optimization to converge and allows a direct comparison with the electric field spatial profile that can be fully estimated in the simulations. For the simulations performed within the optimization procedure, we meshed the geometry sufficiently but keeping the meshing not too fine to limit the computational time required for the algorithm to converge. To again facilitate the convergence of the algorithm, we also smoothed the electric field spatial profile in the simulation using a 200 μm window. We defined a cost function as $f = \Sigma_i \, (y_{i,sim} - y_{i,data})^2 / \Sigma_i \, y_{i,data}^{\,2}$ where the index *i* indicates the electric field estimates at different positions for the model ($y_{sim}$) and the experiments fits ($y_{data}$) starting at the first and ending with the last experimentally measured depths (375 μm and 2500 μm). To minimize the cost function we used the default interior-point method of the *fmincon* function in MATLAB where we set the lower and upper bound for the parameters search. The values of the conductances were bounded by the minimum and maximum values reported by Sim4Life

(https://itis.swiss/virtual-population/tissue-properties/database/low-frequency-conductivity/), which were derived by a detailed literature review. In Fig 4G we also reported the mean and standard deviation of the parameters according to the values reported by Sim4Life. In Table 1 we report the minimum and maximum values that we considered for the optimization. Note that a previous computational study assumed a thickness of 100 μm for the CSF layer in the mouse [64] but this number likely depends on the specific brain area considered [65]. Thus, we considered the CSF thickness a parameter to optimize together with the conductivities and manually set it to potentially vary in a quite large interval for the optimization procedure. The optimization was performed starting from the mean value of the conductivity parameters as reported in the literature and the average of the min and max values set for the CSF layer thickness. The optimization was repeated 10 times to identify the global minimum of the cost function. Once the optimal values of the parameters were found, we re-run the simulation with the optimal set while further increasing the meshing quality to obtain our final estimation of the electric field distribution around and beneath the stimulation electrode (Fig 4H,I).

## Supporting information

**S1 Fig. Schematics of the full fabrication process. A**: PaC deposition. **B**: Spin-coating AZnLOF 2070. **C**: UV Exposure. **D**: Developing. **E**: Gold evaporation. **F**: Lift-off. **G**: Spin-coating AZ 10XT. **H**: UV Exposure. **I**: Developing and reactive ion etching. **J**: Inkjet-printing PEDOT:PSS. **K**: Inkjet-printing SU-8. **L**: Final device.
(TIF)

**S2 Fig. In vivo testing of electric currents beyond the electrochemically stable water window. A**: In vivo electrical impedance spectroscopy measurements before applying the stimulation (gray) after applying a set of stimulation with an amplitude of maximum 100 μA (green) and after a subsequent stimulation set with an amplitude of maximum 400 μA (red). The comparable impedance values following stimulation at 100 μA indicate that the electrode performance is unchanged and also suggest the potential for electrode reusability in other experiments. However, higher current intensities may cause irreversible damage, degrading electrode performance. **B**: Occurrence of bubbles (indicated by the red arrow) after the second set of stimulations. Note that they appeared at the intersection between PEDOT:PSS and the gold connector.
(TIF)

## Acknowledgements

We thank Martin Baca for technical support, Christophe Melon for help setting up the surgical procedures and Fanny Cazettes for helpful comments on the manuscript.

## Author contributions

**Conceptualization:** Rita Matta, Davide Reato, David Moreau, Rodney P. O'Connor.

**Data curation:** Rita Matta, Davide Reato.

**Formal analysis:** Rita Matta, Davide Reato.

**Funding acquisition:** Davide Reato, David Moreau, Rodney P. O'Connor.

**Investigation:** Rita Matta, Davide Reato, Alberto Lombardini.

**Methodology:** Rita Matta, Davide Reato, Alberto Lombardini.

**Project administration:** Rita Matta, Davide Reato, David Moreau, Rodney P. O'Connor.

**Resources:** Rita Matta, Davide Reato, Alberto Lombardini.

**Software:** Davide Reato.

**Supervision:** David Moreau, Rodney P. O'Connor.

**Validation:** Rita Matta, Davide Reato, Alberto Lombardini, David Moreau, Rodney P. O'Connor.

**Visualization:** Rita Matta, Davide Reato.

**Writing – original draft:** Davide Reato.

**Writing – review & editing:** Rita Matta, Davide Reato, Alberto Lombardini, David Moreau, Rodney P. O'Connor.

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
