## [Decision Letter · Decision Letter 0]

14 Nov 2024

PONE-D-24-40579Inkjet-printed transparent electrodes for electrical brain stimulationPLOS ONE

Dear Dr. Reato,

Thank you for submitting your manuscript to PLOS ONE. After careful consideration, we feel that it has merit but does not fully meet PLOS ONE’s publication criteria as it currently stands. Therefore, we invite you to submit a revised version of the manuscript that addresses the points raised during the review process.

We look forward to receiving your revised manuscript.

Kind regards,

Mallikarjuna Reddy Kesama, Ph.D.

Academic Editor

PLOS ONE

Journal Requirements:

3. Thank you for stating the following financial disclosure: [This work was funded by Marie-Curie postdoctoral fellowships (D.R.: HORIZON-MSCA-2021-PF-01 101063075), the European Union’s Horizon 2020 research and innovation programme under grant agreement N°101034324 (D.R.), the French government under the France 2030 investment plan, as part of the Initiative d’Excellence d’Aix-Marseille Université – A*MIDEX  AMX-22-COF-132 (D.R.), the French National Research Agency, through the « Investissements d’Avenir » program (ANR-21-ESRE-0003).]. Please state what role the funders took in the study. If the funders had no role, please state: "The funders had no role in study design, data collection and analysis, decision to publish, or preparation of the manuscript." If this statement is not correct you must amend it as needed. Please include this amended Role of Funder statement in your cover letter; we will change the online submission form on your behalf.

 4. Thank you for stating the following in the Acknowledgments Section of your manuscript: [We thank Martin Baca for technical support and Fanny Cazettes for helpful comments on the manuscript. This work was funded by Marie-Curie postdoctoral fellowships (D.R.:HORIZON-MSCA-2021-PF-01 101063075), the European Union’s Horizon 2020 research and innovation programme under grant agreement N°101034324 (D.R.), the French government underthe France 2030 investment plan, as part of the Initiative d’Excellence d’Aix-Marseille Université – A*MIDEX AMX-22-COF-132 (D.R.), the French National Research Agency, through the «Investissements d’Avenir » program (ANR-21-ESRE-0003)] We note that you have provided funding information that is not currently declared in your Funding Statement. However, funding information should not appear in the Acknowledgments section or other areas of your manuscript. We will only publish funding information present in the Funding Statement section of the online submission form. Please remove any funding-related text from the manuscript and let us know how you would like to update your Funding Statement. Currently, your Funding Statement reads as follows: [This work was funded by Marie-Curie postdoctoral fellowships (D.R.: HORIZON-MSCA-2021-PF-01 101063075), the European Union’s Horizon 2020 research and innovation programme under grant agreement N°101034324 (D.R.), the French government under the France 2030 investment plan, as part of the Initiative d’Excellence d’Aix-Marseille Université – A*MIDEX  AMX-22-COF-132 (D.R.), the French National Research Agency, through the « Investissements d’Avenir » program (ANR-21-ESRE-0003).].  Please include your amended statements within your cover letter; we will change the online submission form on your behalf.

5. In the online submission form, you indicated that your data will be submitted to a repository upon acceptance.  We strongly recommend all authors deposit their data before acceptance, as the process can be lengthy and hold up publication timelines. Please note that, though access restrictions are acceptable now, your entire minimal  dataset will need to be made freely accessible if your manuscript is accepted for publication. This policy applies to all data except where public deposition would breach compliance with the protocol approved by your research ethics board. If you are unable to adhere to our open data policy, please kindly revise your statement to explain your reasoning and we will seek the editor's input on an exemption.

Additional Editor Comments:

The manuscript clearly articulates the objectives, methods, and findings. The technical language is appropriate for a specialized audience, though some sections could benefit from simplification to improve accessibility.

The organization is logical and flows well from introduction through to the discussion. Minor improvements in transition sentences between sections could enhance readability. Figures are informative and generally well-organized. Some legends, such as for Figures 2 and 4, could be more descriptive for readers to quickly interpret the data. The introduction provides solid context for the need for transparent electrodes. However, adding a brief comparison of this approach to similar recent works could strengthen the motivation for the study. Consider briefly explaining terms like “2-photon imaging” and “finite-element modeling” for a broader audience who may not specialize in these methods. The methods section is comprehensive but could be condensed by removing redundancy in protocol descriptions (e.g., the procedure for inkjet printing PEDOT). Consider adding references for well-established protocols rather than describing them in full detail. The manuscript briefly mentions ethical approval for animal studies, which is good. You could add a sentence on efforts to minimize animal suffering and ensure welfare. The results are generally well-presented, but adding statistical markers directly within Figures (like error bars or p-values) could help reinforce findings. In the section describing Electrochemical Impedance Spectroscopy, consider expanding on how the impedance values achieved relate to similar electrode materials in other studies. The discussion could benefit from a paragraph comparing the PEDOT electrode’s transparency, flexibility, and charge capacity to ITO or graphene-based electrodes. The application potential is well-covered, though highlighting potential clinical implications beyond research use could expand the relevance of the work. The limitations of the study could be discussed more explicitly, particularly any known weaknesses of PEDOT in long-term stability. The conclusion effectively summarizes the significance of the findings. A final sentence emphasizing the next steps or long-term goals for this research could leave a stronger impression. These suggestions are intended to help enhance the manuscript's clarity, strengthen the arguments, and ensure that the presentation of the research is as accessible and impactful as possible.

Reviewers' comments:

Reviewer's Responses to Questions

**Comments to the Author**

1. Is the manuscript technically sound, and do the data support the conclusions?

Reviewer #1: Yes

Reviewer #2: Yes

2. Has the statistical analysis been performed appropriately and rigorously? 

Reviewer #1: Yes

Reviewer #2: N/A

3. Have the authors made all data underlying the findings in their manuscript fully available?

Reviewer #1: Yes

Reviewer #2: Yes

4. Is the manuscript presented in an intelligible fashion and written in standard English?

Reviewer #1: Yes

Reviewer #2: Yes

5. Review Comments to the Author

Reviewer #1: Overall, this paper introduced a novel transparent electrode for electrical stimulation, with comprehensive property characterizations. The transparent electrode allows brain imaging tools to monitor the brain activity during the stimulation. I personally like the idea. This paper is high-quality and innovative.

Comments:

1. The PEDOT:PSS electrode presented has a small size with only 1 mm diameter. So, the current density injected can be high. This paper should discuss if the electrode surface is heating up or not, and discuss if it will cause any discomforts to the users if used for person.

2. In terms of the electrode-skin interface, does a skin preparation is required for mouse experiments? For example, shaving the fur on the skin? And if the electrode is used on human head, does the human hair affect the electrode contact quality?

3. Does a conductive gel or electrolyte saline required during the stimulation to improve the electrode impedance?

4. The paper should discuss if the electrode is disposable or can be used repeated.

5. I am just wondering if this electrode can be used for EEG brain monitoring, because its very small size and good electrical performance, if yes, it might be a good solution to overcome the obstacle of the human hair, where most of the on-skin flexible electrode cannot. (as discussed in this paper: doi: 10.1109/FLEPS51544.2021.9469782)

Reviewer #2: The authors report the development and preliminary testing of a electrode for neurostimulation. A hybrid cleanroom dependent and clean room free inkjet printing approach has been used for electrode fabrication. While similar approaches have been used in the past to develop electrodes for neuroscience applications, the manuscript discusses results in a mouse model, which advocates for consideration of this manuscript. The experimentation for use case in animals is limited but the work covers fabrication, and the analysis is quite detail. I particularly commend the discussion of how the 300 V/m electric field compares to the common brain stimulation applications.

I suggest the following to be implemented in the revised manuscript before it can be considered further:

• It should be categorically established how this work compares with existing inkjet printed PEDOT:PSS electrodes for implantable applications. One important reference is https://doi.org/10.1002/pssa.202100683 (Transparent, Low-Impedance Inkjet-Printed PEDOT:PSS Microelectrodes for Multimodal Neuroscience). A benchmarking/comparison table is necessary to bring out novelty/USP of the current work. Also, sensing and transduction should not be the only comparative factor as it’s quite intuitive that in neural and cardiac systems, the dual use capability of electrodes is obvious.

• It is my opinion that the title needs revisiting. There are only primitive studies for deployment in animal models and thus the current form of title is an ambitious representation of the work “Inkjet-printed transparent electrodes for electrical brain stimulation”. I believe making the title modest is necessary to indicate the true nature of the work, something like “Inkjet-printed transparent electrodes: Towards electrical brain stimulation”, or “Promising Approach…”, etc.

• Limitations of the work have not sufficiently been highlighted.

• The abstract is very superficial in it’s current form with electric field value from simulation highlighted. More quantified data should form part of the abstract.

• Page 5, Line 186: “PEDOT:PSS vs Ag/AgCl ([-0.6 0.8] V)” – I believe “[-0.6 0.8] V” is being used to discuss the potential window. I am of the opinion it should be replaced with explanatory representation to improve paper readability.

6. PLOS authors have the option to publish the peer review history of their article (what does this mean?). If published, this will include your full peer review and any attached files.

Reviewer #1: **Yes: **Le Xing

Reviewer #2: No

---

## [Author Response · Author response to Decision Letter 0]

31 Jan 2025

We thank the Editor and the Reviewers for their comments and suggestions. We addressed the points one by one below, including the editorial ones that we split to improve clarity. Thanks to the reviewing feedback, we believe that the manuscript is now largely improved.

Additional Editor Comments:

The manuscript clearly articulates the objectives, methods, and findings. The technical language is appropriate for a specialized audience, though some sections could benefit from simplification to improve accessibility.

We thank the editor for their positive assessment and helpful comments on our manuscript. We address each comment individually below.

1) The organization is logical and flows well from introduction through to the discussion. Minor improvements in transition sentences between sections could enhance readability.

In response to the editor's suggestion to improve flow, we have added transition sentences throughout the manuscript.

We added the following sentence at the beginning of the section titled Electrical characterization (line 166): “We began by measuring the overall thickness of the printed PEDOT:PSS layers, as this variable likely influences both the charge storage capacity of the electrodes (and thus the injectable current) and the device transparency.”

We added the following sentence at the beginning of the section titled Transmittance measurements (line 234): “We characterized the transparency of the devices to find a thickness that balances sufficient current injection with high transparency.”

2) Figures are informative and generally well-organized. Some legends, such as for Figures 2 and 4, could be more descriptive for readers to quickly interpret the data.

We have added sentences to the figure legends to facilitate a clearer and more immediate understanding of the data.

The legend for Figure 2 now reads (new text is underlined):

Figure 2. Electrical properties of inkjet printed electrodes. A: Thickness of electrodes made of a different number of printed PEDOT:PSS layers (n = 5, 3, 5, 2, 2 electrodes for 1-5 layers respectively). B: Electrical impedance spectroscopy measurements for electrodes made of different numbers of inkjet printed PEDOT:PSS layers. Thick lines represent mean values across electrodes and thin lines the error bars estimated as standard deviation (n = 9 electrodes for 1 printed layer and and n = 3 for the others). Due to their low impedance, the electrodes are compatible with commercially available electrical stimulators. C: Cyclic voltammetry (CV) measurements for electrodes made of different numbers of inkjet printed PEDOT:PSS layers. Larger areas enclosed by the cyclic voltammetry curves indicate higher charge storage capacity of the electrodes. D: Charge storage capacity (cathodic: light gray, total: dark gray) estimated from the cyclic voltammetry measurements (calculated from the areas under the cyclic voltammetry curves) for different numbers of layers. E: Example traces (for one and three layers PEDOT:PSS) from pulse tests to estimate the maximum current that can be applied using the electrodes. Dashed lines represent the water window ([-0.6 0.8] V), defining the voltage range where the electrode/electrolyte interface remains stable and avoids significant water electrolysis. F: Maximum current that can be injected using the electrodes before reaching the water window, beyond which undesirable side reactions may occur.

The legend of Figure 4 now reads (new text is underlined):

Figure 4. Combined in vivo experiment and computational modeling to estimate the electric field generated by the inkjet-printed electrode. A: The flexible printed PEDOT:PSS electrode is placed on the mouse brain for stimulation, together with a screw counter electrode in the front to close the circuit. A rigid electrode is used to record voltage fluctuations due to a sinusoidal current stimulation (10 Hz). B: The component of the electric field perpendicular to the brain surface (Ez) is estimated by taking the difference of the amplitude of the recorded sinusoidal voltages at multiple depths (schematic representation). Electric field measurements offer insights into the potential effects of the stimulation at specific spatial locations in the brain. C: Estimated electric field (Ez) for four different currents applied, indicated by different colors. Lines indicate a double exponential fit for each stimulation amplitude. The top plot shows the linearity between current applied and the electric field (both of which are normalized by the values corresponding to the maximum current applied). The black line is the identity line. D: Geometry used for the FEM to simulate the experimental conditions. The stimulating and counter electrodes match the geometry and location of the experiment. E: Representation of the FEM meshing with different colors representing different tissue types (electrode, dura, CSF, brain). Each voxel represents a volume element within the model, where equations governing current flow are solved to estimate the current distribution. F: Experimental data and simulation results for a model in which three parameters (the conductivities of the brain and CSF and the thickness of the CSF layer) were optimized. The model, with optimized parameters, accurately reproduces the experimental data. G: Comparison of the optimal parameters identified with the means and standard deviations derived from multiple studies and reported by Sim4Life (see Materials and Methods). Yellow rectangles represent the minimum and maximum values reported in the literature, which were used as the bounds for the optimization procedure. The CSF thickness varies depending on the location in the brain and therefore we left quite large boundaries for the optimization. The optimized parameters closely match those reported in the literature. H: Predicted electric field beneath and close by the stimulating electrode. Colors indicate electric field magnitude as calculated by the model that best reproduces the experimental data. I: Dependence of the electric field magnitude on the depth directly below the center of the stimulation electrode. The electric field rapidly decays with increasing depth in the brain.

3) The introduction provides solid context for the need for transparent electrodes. However, adding a brief comparison of this approach to similar recent works could strengthen the motivation for the study.

In response to comments from the editor and Reviewers 1 and 2, we have reorganized and expanded the Introduction to better compare our work with previous studies. Specifically, we have added the following text after the section introducing the need for transparent electrodes (line 83): “An alternative strategy is to use nanotubes, nanowires or nanomeshes because of the minimal alteration of the light path offered by these structures [38,39]. However, the biocompatibility of the metals often used in these structures has not yet been fully proven. Fabricating devices using transparent materials represents another option. For instance, indium-tin-oxide (ITO) has been used to fabricate transparent devices. However, this inorganic material is expensive, brittle and limited in terms of flexibility [40]. Graphene is an excellent alternative for transparent electrodes [41–43] but cleanroom fabrication remains less cost-effective. Furthermore, the capacity for charge storage and injection with this material may be restricted [43]. Ideally, electrodes for brain stimulation and concurrent imaging should be highly biocompatible, transparent, and capable of injecting sufficient charge to generate electric fields comparable to those used in established human stimulation protocols.”

We have incorporated the following sentences into the latter part of the Introduction (line 111): “One example is a recent study [55] that proposed directly inkjet printing PEDOT:PSS electrodes onto the scalp of human subjects (rather than pre-manufacturing them [56]) for electroencephalography (EEG) applications, as this could improve electrical contact, which is often affected by the presence of hair [57]. Thin films of PEDOT:PSS can also be highly transparent [58,59]. Notably, a study [60] demonstrated that inkjet-printed PEDOT:PSS electrodes, using a large volume insulated except for a small electrolyte-contact area, can achieve transparency and ideal electrical properties for localized recording or stimulation. This is attributed to the small contact area combined with the material's volumetric capacitance. This work highlights the feasibility of inkjet printing PEDOT:PSS for fabricating electrodes for brain stimulation.”

4) Consider briefly explaining terms like “2-photon imaging” and “finite-element modeling” for a broader audience who may not specialize in these methods.

When introducing 2-photon imaging in the Introduction, we have now included the following parenthetical explanation (line 70): “a microscopy technique capable of imaging deeper into living tissue than other methods”.

When introducing finite-element modeling in the Introduction, we have now included the following parenthetical explanation (line 125): “a computational method for simulating physical phenomena by dividing a complex structure into smaller, simpler elements.”

5) The methods section is comprehensive but could be condensed by removing redundancy in protocol descriptions (e.g., the procedure for inkjet printing PEDOT). Consider adding references for well-established protocols rather than describing them in full detail.

We appreciate the reviewer's feedback regarding the length of the Methods section. We aimed for a comprehensive description of the electrode fabrication process, particularly the inkjet printing of PEDOT:PSS, to ensure reproducibility. Maintaining a detailed protocol is common practice in the field of neural interface fabrication, especially when describing relatively new materials or fabrication techniques, such as those employed in our work. We have carefully scrutinized the Methods section, and we believe that every detail included is crucial for other researchers to accurately replicate our electrode fabrication process. Removing any of these details, in our assessment, would unacceptably hinder the reproducibility of our work. However, to provide further context and background information without adding to the length of the Methods section, we have added a reference to a previous publication that focuses specifically on the ink formulation (line 555): “Further details on the ink formulation rationale and the mechanical and electrical properties of the PEDOT:PSS films are provided in [54].” We hope that this addition, while not reducing the length, helps to address the reviewer’s concern by providing a resource for readers seeking more in-depth information on that particular aspect. We are, of course, open to further suggestions from the editor on this matter.

6) The manuscript briefly mentions ethical approval for animal studies, which is good. You could add a sentence on efforts to minimize animal suffering and ensure welfare.

We modified the following sentence in the Methods section (line 624): “All animal experimental procedures, aimed at minimizing any animal suffering during all surgical procedures, were approved and performed in accordance with the French Ministry of Higher Education, Research and Innovation (approval reference number APAFIS#22182–2019091818381938).”

7) The results are generally well-presented, but adding statistical markers directly within Figures (like error bars or p-values) could help reinforce findings.

Error bars are shown for all data points where more than one sample was available. Statistical analyses, where necessary, are reported directly in the main text.

8) In the section describing Electrochemical Impedance Spectroscopy, consider expanding on how the impedance values achieved relate to similar electrode materials in other studies.

The specific impedance of an electrode is influenced by multiple factors, including its material, geometrical properties, and the interface with the electrolyte. Consequently, direct comparisons of impedance values are challenging unless all these factors are identical. However, we can compare the conductivity of the materials. As indicated in the manuscript, our measured values align with those reported in the paper where the ink formulation was initially introduced. Furthermore, we have added an extra reference, Bihar et al. (2017). Based on the sheet resistance and thickness of the PEDOT:PSS printed electrodes reported in their Supplementary Figure 1, we estimate a conductivity of approximately 100 S/cm. This value is similar to our findings, even though the ink formulation differs slightly. We modified the following sentence to include this reference (line 178): “We performed the measurements using both one layer and two layers. We estimated values of 346 ± 3 Ω/◻ and 174 ± 6 Ω/◻ (“ohms per square”, mean ± SD, n = 7 measurements for one printed layer and n = 6 for two layers), corresponding to conductivity values of 131 ± 1 S/cm and 133 ± 5 S/cm, consistent with values obtained with the same ink formulation [54] as well as with a study reporting approximately 100 S/cm for a similar, though not identical, ink formulation [61].”.

9) The discussion could benefit from a paragraph comparing the PEDOT electrode’s transparency, flexibility, and charge capacity to ITO or graphene-based electrodes.

In the Discussion section, we have added a paragraph that compares the performance and characteristics of ITO, graphene, and PEDOT:PSS. This comparison also includes an analysis of the known limitations of PEDOT:PSS. The new paragraph reads (line 432): “While in our study we used PEDOT:PSS, the predominant materials for transparent electrodes are graphene and sputtered metal oxides, such as ITO. ITO exhibits high transparency (~90%) and excellent conductivity (~10^4 S/cm), with a sheet resistance as low as 36 Ω/◻ [68], but its inherent brittleness limits its flexibility. Conversely, graphene demonstrates over 90% transparency across the ultraviolet to infrared spectrum, coupled with high electrical (10^4 - 10^5 S/cm) and thermal conductivity, as well as flexibility and biocompatibility [69]. Nevertheless, its high cost and complex synthesis processes may limit its broader use for electrode fabrication. PEDOT:PSS, in contrast, distinguishes itself by combining transparency with flexibility [70], while offering mixed electronic and ionic conductivity. Furthermore, it benefits from commercial availability and ease of processing and manipulation [48,71]. Despite its significant promise, PEDOT:PSS presents certain limitations, including restricted conductivity, flexibility, and mechanical stability [72]. The material's complex structure, which involves both electronic and ionic transport, makes its properties highly sensitive to processing techniques. Furthermore, the mechanical stability of PEDOT:PSS can be problematic, with potential issues such as cracking, delamination, and re-dispersion in aqueous environment [73]. While incorporating cross-linkers can enhance film stability, this often comes at the cost of reduced electronic conductivity [74].”

10) The application potential is well-covered, though highlighting potential clinical implications beyond research use could expand the relevance of the work.

We now modified a paragraph in the discussion to highlight potential clinical implications of our work. Specifically, the paragraph now reads (line 513): “While further testing is required to confirm long-term biocompatibility and functionality, our device represents a significant advancement due to its cost-effectiveness and rapid testability. This makes it valuable not only for research but also for clinical applications requiring effective stimulation or recording, as well as optical access or imperceptibility [111]. Transparent PEDOT:PSS electrodes, for instance, are also used in biosensors and wearable health monitoring devices, offering conductivity without sacrificing flexibility or transparency [112]. Furthermore, advanced transparent systems like Carbon Layered Electrode Arrays (CLEAR), composed of graphene and Parylene C, have successfully integrated neuro-electric

---

## [Decision Letter · Decision Letter 1]

18 Feb 2025

Inkjet-printed transparent electrodes: design, characterization, and initial in vivo evaluation for brain stimulation

PONE-D-24-40579R1

Dear Dr. Davide Reato,

We’re pleased to inform you that your manuscript has been judged scientifically suitable for publication and will be formally accepted for publication once it meets all outstanding technical requirements.

Kind regards,

Mallikarjuna Reddy Kesama, Ph.D.

Academic Editor

PLOS ONE

Additional Editor Comments (optional):

Reviewers' comments:

Reviewer's Responses to Questions

**Comments to the Author**

1. If the authors have adequately addressed your comments raised in a previous round of review and you feel that this manuscript is now acceptable for publication, you may indicate that here to bypass the “Comments to the Author” section, enter your conflict of interest statement in the “Confidential to Editor” section, and submit your "Accept" recommendation.

Reviewer #1: All comments have been addressed

Reviewer #2: All comments have been addressed

2. Is the manuscript technically sound, and do the data support the conclusions?

Reviewer #1: Yes

Reviewer #2: Yes

3. Has the statistical analysis been performed appropriately and rigorously? 

Reviewer #1: Yes

Reviewer #2: Yes

4. Have the authors made all data underlying the findings in their manuscript fully available?

Reviewer #1: Yes

Reviewer #2: Yes

5. Is the manuscript presented in an intelligible fashion and written in standard English?

Reviewer #1: Yes

Reviewer #2: Yes

6. Review Comments to the Author

Reviewer #1: Overall, the revised paper has resolved my comments/questions in a clear way. I understand it is challenging to do more practical experiments to answer some of the reviewers' comments, but the authors have added more clarifications/explanations on the manuscript, which makes the paper more clear and rigorous. The replies also answer some of the questions clearly. I am satified with the revision, and my suggestion is ACCEPT. Overall, this paper is innovative in electrode development area.

Reviewer #2: The authors have addressed the comments satisfactorily. The quality of the publication has improved post incorporation of comments from the other reviewer and the editorial review as well.

7. PLOS authors have the option to publish the peer review history of their article (what does this mean?). If published, this will include your full peer review and any attached files.

Reviewer #1: **Yes: **Le Xing

Reviewer #2: No

---

## [Editor Report · Acceptance letter]

PONE-D-24-40579R1

PLOS ONE

Dear Dr. Reato,

I'm pleased to inform you that your manuscript has been deemed suitable for publication in PLOS ONE. Congratulations! Your manuscript is now being handed over to our production team.

Kind regards,

on behalf of

Dr. Mallikarjuna Reddy Kesama

Academic Editor

PLOS ONE